

# Flux-tunable Kitaev chain in a quantum dot array

Juan Daniel Torres Luna[1*], A. Mert Bozkurt[1,2],
Michael Wimmer[1,2] and Chun-Xiao Liu[1,2†]

**1** QuTech, Delft University of Technology, Delft 2600 GA, The Netherlands
**2** Kavli Institute of Nanoscience, Delft University of Technology,
2600 GA Delft, The Netherlands

⋆ jd.torres1595@gmail.com , † chunxiaoliu62@gmail.com

## Abstract

Connecting quantum dots through Andreev bound states in a semiconductor-superconductor hybrid provides a platform to create a Kitaev chain. Interestingly, in a double quantum dot, a pair of poor man's Majorana zero modes can emerge when the system is fine-tuned to a sweet spot, where superconducting and normal couplings are equal in magnitude. Control of the Andreev bound states is crucial for achieving this, usually implemented by varying its chemical potential. In this work, we propose using Andreev bound states in a short Josephson junction to mediate both types of couplings, with the ratio tunable by the phase difference across the junction. Now a minimal Kitaev chain can be easily tuned into the strong coupling regime by varying the phase and junction asymmetry, even without changing the dot-hybrid coupling strength. Furthermore, we identify an optimal sweet spot at $\pi$ phase, enhancing the excitation gap and robustness against phase fluctuations. Our proposal introduces a new device platform and a new tuning method for realizing quantum-dot-based Kitaev chains.

| | |
|---|---|
| Received | 21-02-2024 |
| Accepted | 28-08-2024 |
| Published | 18-09-2024 |

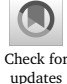
# 1 Introduction

Recently, double quantum dots connected by a superconducting segment have emerged as a promising platform for implementing high-performance Cooper pair splitters [1–3] and for exploring Majorana physics in the Kitaev chain [4–8]. A crucial idea is the use of Andreev bound state (ABS) in a semiconductor-superconductor hybrid to mediate both elastic cotunneling (ECT) and crossed Andreev reflection (CAR) between quantum dots [9]. Moreover, their relative amplitude can be precisely controlled by varying the chemical potential in the hybrid region [10]. This control allows for fine-tuning ECT and CAR strengths and enabling the creation of a pair of Majorana zero modes localized at the outer dots [6–8]. These zero modes are exotic quasiparticle excitations that obey non-Abelian statistics, serving as the building block for implementing topological quantum computing [11–25]. An extra advantage of using quantum dots [26–28] as the material platform for realizing Majorana zero modes is their intrinsic robustness against the effect of disorder in hybrid nanostructures [4].

On the other hand, ABSs are ubiquitously present in Josephson junctions, investigated extensively in various fields such as Andreev spin qubits [29–32], superconducting diode effect [33–38], and topological superconductivity [39–43] with potential applications in quantum technologies. A common aspect in these applications involves manipulating ABS properties, such as energy and wavefunctions, through control of the phase difference across the Josephson junction. This phase difference, dependent on the magnetic flux threaded through a superconducting loop, provides a distinct advantage over conventional methods of controlling ABS properties using electrostatic gate voltages. The flux control method eliminates challenges associated with cross-talk often encountered in electrostatic gate voltage control.

In this work, we theoretically study how to create a minimal Kitaev chain using ABS in a short Josephson junction. In particular, we demonstrate that the phase difference across the Josephson junction not only changes the energy of the ABS, but also varies the BCS charge, determining the relative amplitude of ECT and CAR couplings. Consequently, by varying the magnetic flux threading through the superconducting loop, a sweet spot condition can be reached, leading to the emergence of a pair of poor man's Majorana zero modes. Our proposal therefore opens up a new way for realizing a Kitaev chain with a short Josephson junction using superconducting phase difference as a tuning knob.

The rest of this work is organized as follows: After introducing the model and Hamiltonian in Sec. 2, we analyze the CAR and ECT couplings using perturbation theory in Sec. 3. In Sec. 4 we perform a numerical study of the full many-body Hamiltonian to demonstrate the presence of poor man's Majorana zero modes, and further optimize the sweet spot. Section 5 is devoted to conclusion and discussion.

# 2 Model and Hamiltonian

The system consists of double quantum dots connected by an Andreev bound state in a short Josephson junction, as shown in Fig. 1(a). The full many-body Hamiltonian is

$$H = H_D + H_A + H_{\text{tunn}}, \tag{1}$$

$$H_D = \sum_{i=L,R} \sum_{\sigma=\uparrow,\downarrow} (\mu_i + \sigma E_Z) n_{i\sigma} + U n_{i\uparrow} n_{i\downarrow}, \tag{2}$$

$$H_A = \mu_M \sum_\sigma n_{M\sigma} + [\Gamma_+ \cos(\phi/2) + i\Gamma_- \sin(\phi/2)] c_{M\uparrow}^\dagger c_{M\downarrow}^\dagger + \text{h.c.}, \tag{3}$$

$$H_{\text{tunn}} = \sum_\sigma t \left( c_{L\sigma}^\dagger c_{M,\sigma} + c_{R\sigma}^\dagger c_{M\sigma} \right) + t_{\text{so}} \left( c_{L\sigma}^\dagger c_{M\bar\sigma} - c_{R\sigma}^\dagger c_{M\bar\sigma} \right) + \text{h.c.} \tag{4}$$

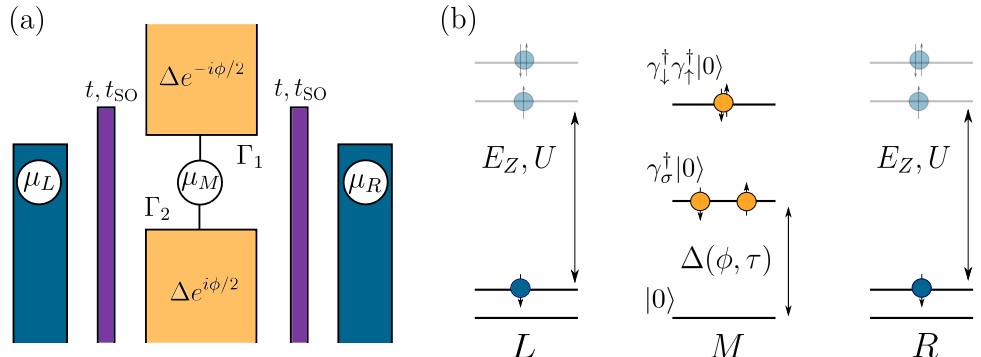

Figure 1: Minimal Kitaev chain with control of the superconducting phase difference. The scheme of the device in panel (a) shows a three-dot chain where the middle dot is connected to two superconducting leads (orange) with couplings $\Gamma_1$ and $\Gamma_2$ and with a phase difference $\phi$. Electrostatically defined quantum dots (blue) are connected to the middle dot by tunnel gates (purple). The energy levels of each dot are shown in panel (b). Outer dots are in the spinless limit (blue dots), *i.e.* $E_Z, U \gg \Delta$, where higher levels (lower transparency) are not relevant. Middle proximitized quantum dot hosts Andreev bound state excitations given by Eq. (7) and depicted as orange dots separated by a gap given in Eq. (6). Here $|0\rangle$ is the singlet ground state of the ABS.

Here, $H_D$ is the Hamiltonian describing the two outer quantum dots, $n_{i\sigma} = c_{i\sigma}^\dagger c_{i\sigma}$ is the electron number operator, $\mu_i$ is the dot energy, $E_Z$ is the induced Zeeman spin-splitting energy, and $U$ is the dot Coulomb interaction. $H_A$ is the Hamiltonian for the ABS forming in a Josephson junction, which is effectively described by a quantum dot weakly coupled to two superconducting leads [44]. $\mu_M$ is the chemical potential of the middle dot, $\Gamma_\pm = (\Gamma_1 \pm \Gamma_2)/2$ are the symmetrized and anti-symmetrized superconducting couplings with $\Gamma_1, \Gamma_2 > 0$, and $\phi$ is the superconducting phase difference. $H_{\text{tunn}}$ describes the tunneling between the quantum dots and the ABS, where $t$ is the amplitude for spin-conserving processes, while $t_{\text{SO}}$ is for spin-flipping ones due to spin-orbit interaction in the system. Without loss of generality, we perform a gauge transformation on the pairing terms to make $H_A$ real, *i.e.*, $c \to \tilde{c} = e^{i\theta/2}c$ with $\theta = \arctan\left[\frac{\Gamma_+}{\Gamma_-}\tan(\phi/2)\right]$. As a result, the ABS Hamiltonian becomes

$$H_A = \mu_M \sum_\sigma \tilde{c}_{M\sigma}^\dagger \tilde{c}_{M\sigma} + \Delta(\phi, \tau)\tilde{c}_{M\uparrow}^\dagger \tilde{c}_{M\downarrow}^\dagger + \text{h.c.}, \tag{5}$$

$$\Delta(\phi, \tau) = \sqrt{\Gamma_+^2 \cos^2(\phi/2) + \Gamma_-^2 \sin^2(\phi/2)} = \Gamma_+\sqrt{1 - \tau \sin^2(\phi/2)}, \tag{6}$$

where $\Delta(\phi, \tau)$ is the phase-dependent the effective superconducting gap and $\tau = (\Gamma_+^2 - \Gamma_-^2)/\Gamma_+^2$ is the effective transparency of the Josephson junction. In what follows, we choose $\Gamma_+ = 1$ as the natural unit in our calculations and characterize the junction asymmetry using $\eta = \Gamma_-/\Gamma_+$.

In order to obtain more insight into the system, we diagonalize the ABS Hamiltonian and obtain the ABS energy. The excitations in the ABS correspond to the operators

$$\gamma_\sigma = u c_{M\sigma} + v c_{M\bar{\sigma}}^\dagger, \qquad \gamma_\sigma^\dagger = u c_{M\sigma}^\dagger - v c_{M\bar{\sigma}}, \tag{7}$$

where $u$ and $v$ are the coherence factors of the ABS. The Hamiltonian from Eq. (5) in this basis is $H_A = \sum_\sigma \epsilon_{\text{ABS}} \gamma_\sigma^\dagger \gamma_\sigma$, where the ABS energy is

$$\epsilon_{\text{ABS}}(\phi, \tau) = \sqrt{\mu_M^2 + [\Delta(\phi, \tau)]^2}. \tag{8}$$

In the excitation basis, the gap between the ground state and the excited states depends on the phase difference $\phi$ and the junction asymmetry $\eta$ which is depicted in Fig. 1(b).

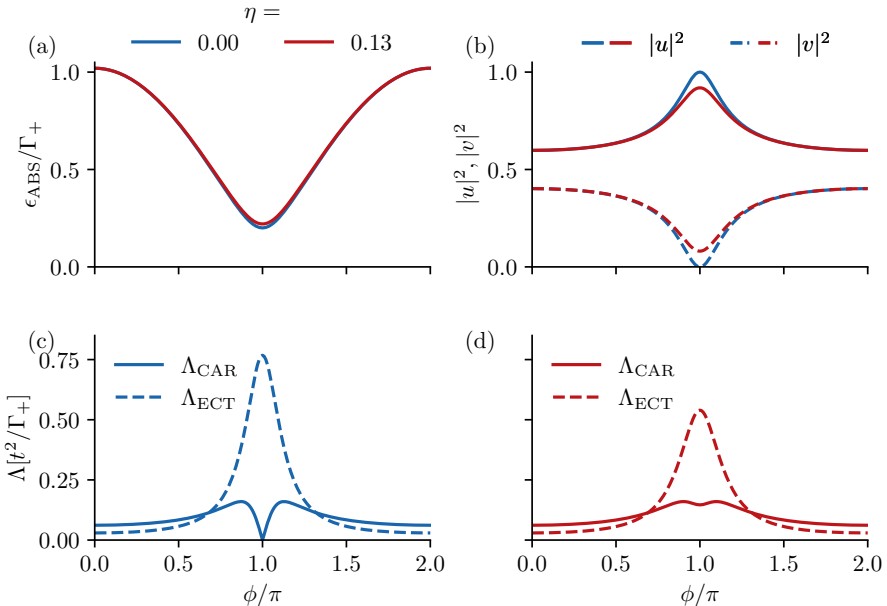

Figure 2: ABS energy (a) from Eq. (8) and magnitude of electron-hole components (b) from Eq. (7) for two different junction asymmetries $\eta$ as a function of phase difference $\phi$. Strength of effective ECT [Eq. (11)] and CAR [Eq. (10)] as a function of $\phi$ for symmetric (c) and asymmetric (d) junctions from the effective model defined in Eq. (9). We set $\mu_M = 0.2$, and $t_{\mathrm{SO}} = 0.2 \times t$.

## 3 Flux-tunable CAR and ECT

To gain insight into how the ABS in a short Josephson junction mediates couplings between outer quantum dots, we first focus on the tunneling regime, *i.e.*, $t, t_{so} \ll \Gamma_+$, and calculate the CAR and ECT amplitudes using second-order perturbation theory. We further assume strong Zeeman spin splitting in quantum dots ($t, t_{so} \ll E_Z$), such that we only need to consider one spin species, and for concreteness but without loss of generality, we consider the scenario of spin up for both quantum dots. Moreover, we assume strong local charging energy in the outer quantum dots as reported in recent experiments [7, 8].[1] Under these assumptions, by performing a Schrieffer-Wolff transformation we obtain the effective couplings of the outer dots as below

$$H_{\mathrm{eff}} = \Lambda_{\mathrm{CAR}}(\phi, \tau) c_{L\uparrow}^\dagger c_{R\uparrow}^\dagger + \Lambda_{\mathrm{ECT}}(\phi, \tau) c_{L\uparrow}^\dagger c_{R\uparrow} + \mathrm{h.c.}, \tag{9}$$

$$\Lambda_{\mathrm{CAR}}(\phi, \tau) = 2 t\, t_{\mathrm{so}} \cdot \frac{\Delta(\phi, \tau)}{\mu_M^2 + \Delta^2(\phi, \tau)}, \tag{10}$$

$$\Lambda_{\mathrm{ECT}}(\phi, \tau) = (t_{\mathrm{so}}^2 - t^2) \cdot \frac{\mu_M}{\mu_M^2 + \Delta^2(\phi, \tau)}, \tag{11}$$

where $\Lambda_{\mathrm{CAR}} \propto 2uv$ is the CAR amplitude, which physically is given by coherent tunneling of an electron followed by a hole (or vice versa), while $\Lambda_{\mathrm{ECT}} \propto (u^2 - v^2)$ is the ECT amplitude mediated by tunneling events of two electrons or two holes [9]. We validate the results from perturbation theory with the full many-body Hamiltonian in Appendix B. A key observation here is that the phase difference not only varies the energy of the ABS [see Fig. 2(a)], but

---

[1]The behavior of the system in the large charging energy limit would be similar to the case with vanishing charging energy given that the Zeeman splitting in the quantum dots is sufficiently large.

also changes its electron and hole components [see Fig. 2(b)], therefore giving a different dependence of ECT and CAR amplitudes on the phase difference. In particular, for a symmetric junction [see Fig. 2(c)], in the vicinity of $\phi = 0$ and $2\pi$, the BCS charge of the ABS is closest to neutrality, leading to a stronger CAR amplitude compared to ECT. In contrast, at $\phi = \pi$, due to the destructive interference of the two superconducting leads, the ABS becomes totally electron-like ($u^2 = 1$), suppressing the CAR amplitude while maximizing the ECT amplitude, as shown in Fig. 2(c). Even when the junction is not symmetric ($\eta = 0.13$) as in the idealized case , the qualitative features found in the CAR and ECT curves remain the same [see Fig. 2(d)], except that now the dip of CAR curve at $\phi = \pi$ is lifted owing to imperfect destructive interference. The crucial finding in our analytical results of CAR and ECT amplitudes is that as the phase difference varies from 0 to $\pi$, their ratio goes from $\Lambda_{\mathrm{ECT}}/\Lambda_{\mathrm{CAR}} \ll 1$ to $\Lambda_{\mathrm{ECT}}/\Lambda_{\mathrm{CAR}} \gg 1$. This guarantees the presence of a particular value of $\phi$ where CAR and ECT amplitudes become equal. In essence, we demonstrate that adjusting the phase difference across the Josephson junction effectively controls the coupling between two quantum dots, creating a tunable minimal Kitaev chain.

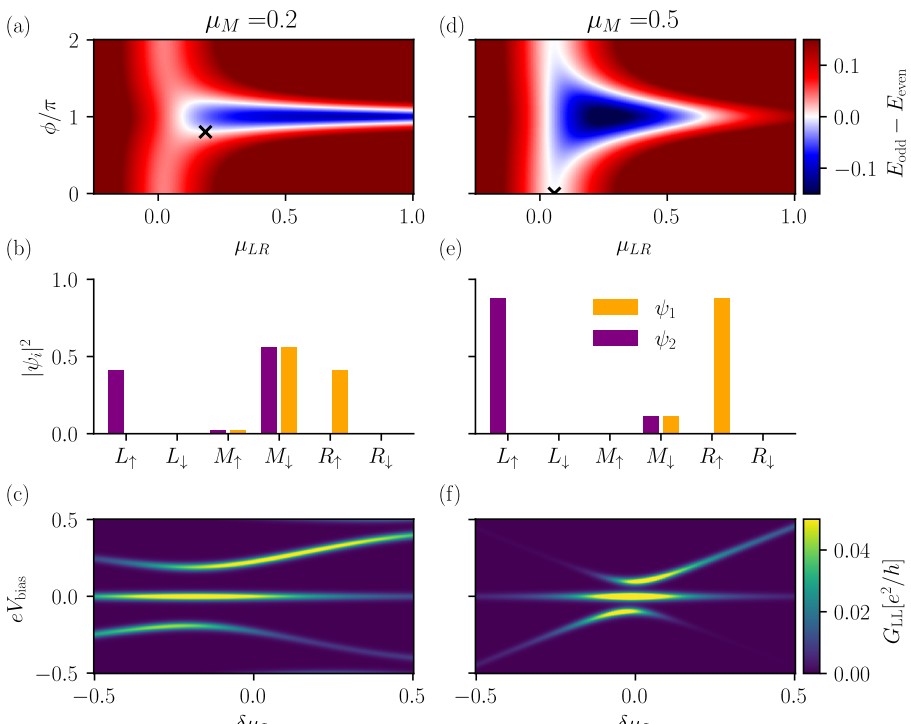

Figure 3: Sweet spots for different middle chemical potentials $\mu_M$ at $\tau = 1$. (a, d) Phase diagram of Eq. (1) as a function of $\phi$ and $\mu_{LR} = \mu_L = \mu_R$ for two different $\mu_M$ indicated at the top. At the bottom of the degeneracy line, a black cross that indicates the sweet spot. (b, e) Majorana wave functions are calculated at the sweet spots shown in the above panels. Local conductance as a function bias voltage $V_{\mathrm{bias}}$ and detuning of the left chemical potential $\delta\mu_L$ for the corresponding sweet spots (c, f). The temperature of the normal leads is $T = 0.01$, and the lead-dot coupling strength is $\Gamma_{LD} = 0.01$. Other parameters are $E_Z = 4$ and $U = 2$.

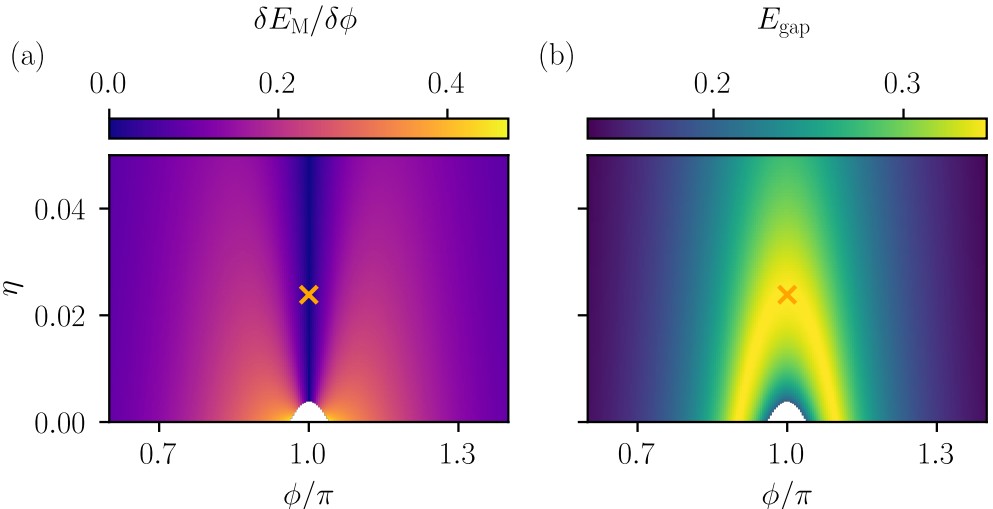

Figure 4: Sweet spot quality as a function of junction transparency $\eta$ and phase difference $\phi$. Blank regions represent the points where there is no sweet spot. Deviation of the Majorana energy $E_M$ with respect to phase fluctuation is shown in panel (a). The excitation gap $E_{\text{gap}}$ for each sweet spot is shown in panel (b). For each pair $(\phi, \eta)$, the chemical potentials $\mu_L = \mu_R$ and $\mu_M$ are tuned to the corresponding sweet spot.

## 4 Numerical study of the full many-body Hamiltonian

We extend our analysis beyond perturbation theory, considering the full many-body Hamiltonian in Eq. (1), incorporating realistic effects in an experimental setup such as the finite strength of Zeeman spin splitting and Coulomb interaction in quantum dots. We apply the exact diagonalization method to calculate the ground-state phase diagram and the Majorana wavefunctions. Furthermore, we perform transport simulations using the rate equation formalism [45, 46] and calculate the local conductance spectroscopy.[2] In particular, we consider leads at a finite temperature $T$ coupled to the outer dots with a coupling strength $\Gamma_{LD}$. Figure 3(a) illustrates the phase diagram, defined as the energy difference between ground states with opposite fermion parity, in the $(\mu_{LR}, \phi)$ plane for system parameters $\mu_M = 0.2$, $E_Z = 4$ and $U = 2$. A stark feature is that a blue region describing $E_{\text{odd}} < E_{\text{even}}$ appears in the vicinity of $\phi = \pi$, contrasting with the red regions where $E_{\text{odd}} > E_{\text{even}}$. The phase diagram thus indicates that ECT couplings dominate around $\pi$ phase, consistent with our analytic results based on perturbation theory, as shown in Fig. 2(c). Moreover, the white line, denoting the degeneracy between even- and odd-parity ground states, has a critical point at the bottom indicated by a black cross, of which the tangent line is parallel to the $x$-axis. We define this point as the sweet spot of the coupled quantum dots, hosting a pair of poor man's Majoranas. As depicted in Fig. 3(b), the wavefunctions of the two Majoranas are decoupled. Namely, the first Majorana wavefunction is localized in the left dot and the second one in the right dot. We emphasize here that wavefunction overlap in the middle ABS is not detrimental in assessing the quality of Majorana zero modes, because the ABS works as a virtual coupler in this setup. Additionally, the calculated conductance spectroscopy shown in Fig. 3(c) further confirms that a Majorana-induced zero-bias conductance peak is robust against detuning of a single dot chemical potential ($\delta \mu_L$).

---

[2]The rate equation formalism does not capture coherent co-tunneling processes.

For comparison, we also consider a scenario with a different value of middle dot chemical potential $\mu_M = 0.5$ (see the left column in Fig. 3). The qualitative features of the phase diagram are quite similar to those of $\mu_M = 0.2$ except that now the sweet spot appears close to $\phi = 0$ [see Fig. 3(b)], where the ABS energy is the highest in the phase sweep [see Fig. 2(a)]. As a result, even without changing the strength of the dot-hybrid coupling, the system now enters the weak-coupling regime [46], giving negligible wavefunction leakage into the ABS [see Fig. 3(d)] and a smaller excitation gap [see Fig. 3(f)].

Owing to the fact that Majoranas for different system parameters can have different physical properties, e.g., excitation gap size, we investigate the optimal sweet spot condition in double quantum dots by varying the three experimentally relevant parameters: middle dot chemical potential $\mu_M$, junction asymmetry $\eta$ and superconducting phase $\phi$. We note that only two parameters are free, with the third parameter depending on the other two in order to obtain a sweet spot. To visualize the impact of the experimentally relevant parameters, Figure 4 illustrates Majorana properties in the $(\phi, \eta)$ plane, emphasizing the roles of junction asymmetry and superconducting phase difference. In Fig. 4 every point corresponds to the sweet spot identified at the bottom of the degeneracy line in the phase diagram as in Fig.3(a, d). It is not always possible to tune into a sweet spot because when the superconducting phase is around $\pi$ for a nearly symmetric Josephson junction, the ABS becomes gapless, making the physical mechanism of using ABS as a coupler break down. Nevertheless, outside those regions in parameter space, we are able to characterize the quality of the Majorana excitations by computing the variation of the ground state energy splitting $E_M \equiv E_{\text{odd,gs}} - E_{\text{even,gs}}$ with respect to phase $\phi$ and the excitation gap $E_{\text{gap}}$. Figure 4(a) shows that for sweet spots located at $\phi = \pi$, the Majorana energy is insensitive to $\phi$ fluctuations up to the first order for all $\eta$. Fig. 4(b) shows that along a ring of low junction asymmetry and close to $\pi$ phase, the excitation gap is maximized. Combining these two findings, we conclude that there is an optimal sweet spot that occurs when the phase is $\phi = \pi$ and the junction symmetry is high but not perfect, *i.e.* the junction is almost transparent (see the cross in Fig. 4). At this point, the poor man's Majoranas are most robust against phase fluctuations and simultaneously possess the largest excitation gap.

## 5 Conclusions and outlook

In this work, we showed that ABS in a short Josephson junction can mediate CAR and ECT couplings between quantum dots, with their relative amplitude being tunable by a novel knob, namely, the superconducting phase difference. The tunability of the system makes it a very suitable platform for creating Kitaev chains and fine-tuned Majorana zero modes. Moreover, we find that the sweet spot is optimized when the superconducting phase difference is close to $\pi$ and the junction is slightly asymmetric. In this optimal regime, the Majorana zero energy becomes insensitive to phase fluctuations and the excitation gap is maximized. Interestingly, a larger excitation gap can be easily reached by varying the superconducting phase and junction asymmetry, eliminating the need to change the dot-hybrid coupling strength via tunnel gate voltage, as demonstrated in prior research [46].

Motivated by developments in semiconductor-superconductor hybrid devices [1–3, 9, 10], we propose extending the two site flux-tunable Kitaev chain into a longer chain in a two-dimensional electron gas as shown in Fig. 5. Our proposal is within experimental reach because it is based on flux-control of a single Andreev state, which is ubiquitous in superconducting devices [47, 48]. Therefore our work opens up a new platform for realizing longer Kitaev chains with high-quality Majorana excitations and an enhanced gap by utilizing a fundamental property of Andreev bound states, namely flux-control, as a tuning knob for the Kitaev chain.



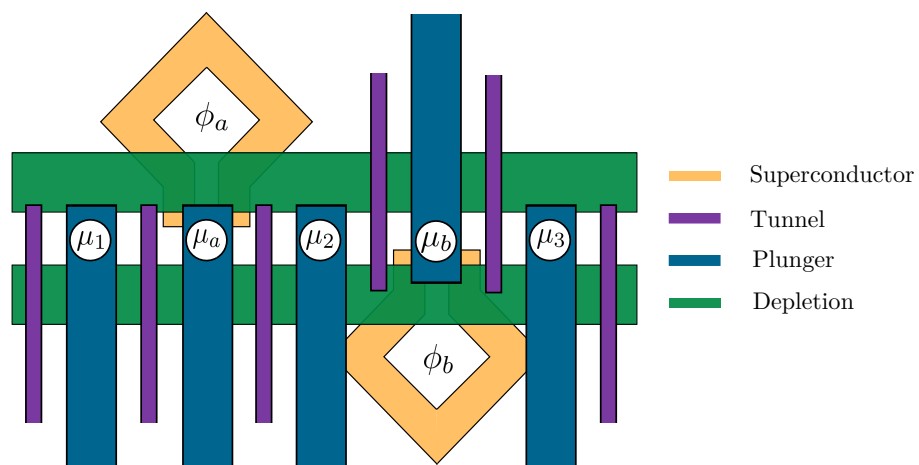

Figure 5: Device proposal for a multi-site flux-tunable Kitaev chain in a 2DEG. The device is defined by multiple layers of gates and one layer of superconducting material separated by dielectric material. The colors of each layer are defined in the legend. The normal quantum dots have a chemical potential $\mu_i$ where $i = 1, 2, \ldots$ The ABS dots are proximitized by two superconductors with phase difference $\phi_j$ and have a chemical potential $\mu_j$ where $j = a, b, \ldots$

## Acknowledgments

We are grateful to I. Kulesh, S.L.D. ten Haaf, S. Roelofs, F. Zatelli, and S. Goswami for numerous useful discussions on experimental devices. We thank A. R. Akhmerov and K. Vilkelis for helpful discussions on numerical simulations.

**Author contributions** C.-X.L. proposed the idea and designed the project. J.D.T.L. performed the analytical calculations with input from C.-X.L., and carried out the numerical simulations with input from A.M.B. J.D.T.L. generated all the figures. C.-X.L. and M.W. supervised the project. All authors discussed the results and contributed to writing of the manuscript.

**Data availability** All the code and the data used in this manuscript are in Ref. [49].

**Funding information** This work was supported by a subsidy for top consortia for knowledge and innovation (TKl toeslag), by the Dutch Organization for Scientific Research (NWO) through OCENW.GROOT.2019.004, by the European Union's Horizon 2020 research and innovation programme FET-Open Grant No. 828948 (AndQC), and by Microsoft Corporation.

## A Majorana wavefunctions and polarization

In this appendix, we present the Majorana wavefunction and Majorana polarization of a minimal Kitaev chain. Because the quantum dots are weakly coupled to the ABS, we approximate the global Majorana operator as a sum of all local Majorana operators:

$$\gamma_{1,j\sigma}(\theta) = e^{i\theta} c_{j\sigma}^{\dagger} + e^{-i\theta} c_{j\sigma}, \tag{A.1}$$

$$\gamma_{2,j\sigma}(\theta) = i\left(e^{i\theta} c_{j\sigma}^{\dagger} - e^{-i\theta} c_{j\sigma}\right), \tag{A.2}$$

where $j = L, M, R$ is the index of the quantum dot, $\sigma$ is the spin index and $\theta$ is a global gauge. We characterize the zero-energy excitation of the minimal Kitaev chain by computing the Majorana wavefunctions. We define the Majorana wavefunction in a given quantum dot as the matrix element, $\psi_{i,j}(\theta) = \sum_\sigma |\langle e|\gamma_{i,j\sigma}(\theta)|o\rangle|^2$, where $|e\rangle$ and $|o\rangle$ are the even and odd many-body ground states, $i = 1, 2$ is the index of the Majorana operator. We interpret this matrix element as the wavefunctions of the zero-energy excitations in the many-body ground state, albeit being gauge-dependent. Based on this definition, we can calculate the Majorana polarization at the $i$-th quantum dot as

$$\mathcal{M}_i(\theta) = \frac{\psi_{1,i}(\theta) - \psi_{2,i}(\theta)}{\psi_{1,i}(\theta) + \psi_{2,i}(\theta)}. \tag{A.3}$$

As the Majorana operators are gauge dependent, the wavefunctions, and therefore the polarization, are also gauge dependent. In Fig. 6(a) we show the Majorana polarization of one of the Majorana wavefunctions in the left dot, as a function of the superconducting phase difference $\phi$ and gauge $\theta$ defined in Eq. (A.1). In Fig. 6(b) we show that the localization of the Majorana wavefunction depends on the gauge choice. The Majorana polarization is unity for the specific gauge [see black line in Fig. 6(a)] that makes the Hamiltonian real, which in turn corresponds to Majorana wavefunctions being maximally localized in the outer quantum dots.

## B   Benchmark of perturbative result

When the coupling between quantum dots and ABS is sufficiently weak, $t, t_{\text{SO}} \ll \Gamma_+$, one can project the system onto a low energy effective Hamiltonian that only describes the outer dots as described in Eq. (9). Using second order perturbation theory, we find that the dominant couplings between the outer dots are the effective CAR and ECT couplings as described in Eq. (10) and Eq. (11). In order to validate the perturbative results, we perform exact diagonalization of the full many-body Hamiltonian in Eq. (1) and extract the effective CAR and ECT couplings following the method described in Ref. [46]. We look at the excitation spectrum at the center of the anti-crossing in the charge stability diagram where the energy splitting between even

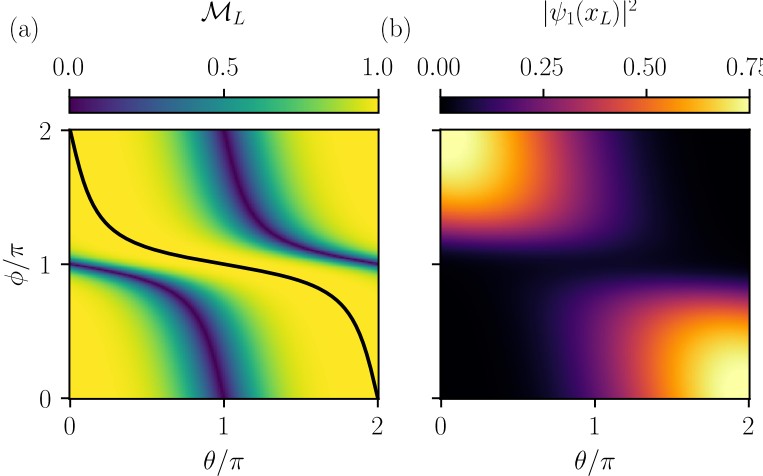

Figure 6: Gauge dependence of the polarization (a) and the Majorana wavefunction in the left dot (b) as a function of phase difference $\phi$. We fix the junction transparency to $\eta = 0.1$ and tune the system into the sweet spot for every $\phi$. The black line in panel (a) corresponds to the gauge choice of a real Hamiltonian.

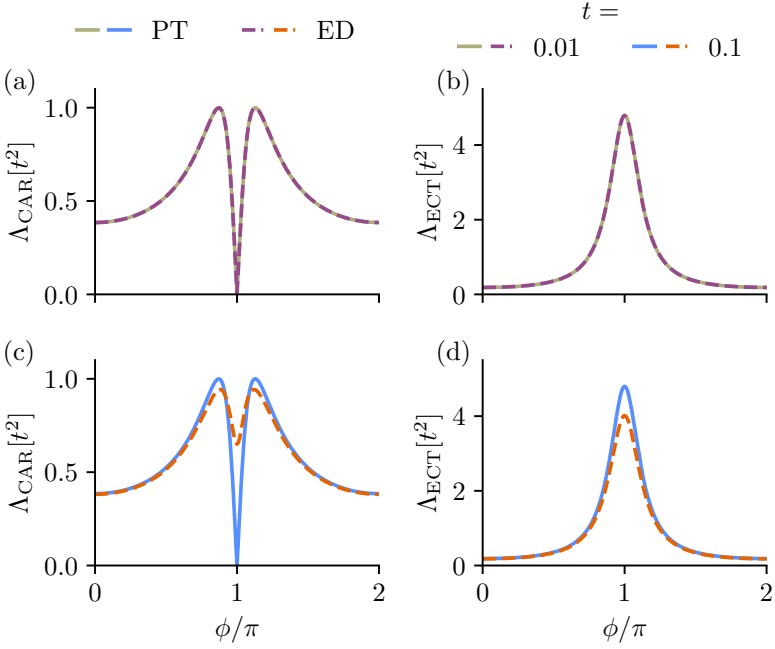

Figure 7: Comparison of CAR and ECT from perturbation theory (solid lines) given by Eq. (10) and Eq. (11) and exact diagonalization (dashed lines) for a small coupling (a, b) and larger coupling (c, d). This plot shows results for perfect transmission $\eta = 0$.

and odd ground states yields $\Lambda_{\text{ECT}} - \Lambda_{\text{CAR}}$, while the excitation gap gives $\Lambda_{\text{ECT}} + \Lambda_{\text{CAR}}$. We compare the results from perturbation theory with the exact diagonalization results in Fig. 7 (a) and (b) where we find good agreement in the weak tunneling regime.

As the coupling between the outer dots and the ABS increases, the perturbative results become less accurate as shown in Fig. 7 (c) and (d). In particular, for $\phi \sim \pi$ perturbative results predict a vanishing CAR amplitude, while the exact diagonalization results show a finite value. We attribute this finite CAR to higher-order Andreev processes that are not captured by the second-order perturbative expansion. Furthermore, higher-order processes become more important as the energy gap in the ABS excitation spectrum decreases. Nevertheless, despite the mismatch between perturbative and exact diagonalization results, one can always find a sweet spot where CAR and ECT are equal.

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
