# Peer review of "Flux-tunable Kitaev chain in a quantum dot array"

_SciPost Physics, doi:SciPost Phys. Core 7, 065 (2024)_

## Round 1 · Referee Report · Anonymous (Referee 1) · 2024-3-18

Report
The paper presents an investigation into a three-quantum-dot setup resembling a Josephson junction. The authors explore various aspects such as the impact of phase difference ($\phi$) and coupling strengths ($\Gamma_{1,2}$) on the system's behavior, with a particular focus on the equality of the CAR and ECT amplitudes at a specific value of $\phi$. The presentation is generally clear, and the paper adequately references and cites relevant literature. However, several concerns and areas for improvement have been identified:
-
Inconsistency between Fig. 1-b and the Hamiltonian defined in Eq. 3, Eq. 5, or Eq. 8. The schematic in Fig. 1-b, particularly the orange representation, appears to deviate from the described Hamiltonian formulations.
-
Ambiguities in Fig. 2. It would enhance clarity if the authors compared the energy spectrum and electron-hole components (u, v) of Eq. 5 with the effective Hamiltonian (Eq. 8) and included these comparisons in panels a and b, respectively. Currently, panels a and b seem to derive from Eq. 5, while panels c and d derive from the effective Hamiltonian, which could be confusing for readers.
-
The effective Hamiltonian (Eq. 8) should be elaborated with details upon to ensure self-consistency, especially considering its role in benchmarking with the numerical Exact Diagonalization (ED) results.
-
The theory developed by the authors regarding the limit of large U in the outer dots appears to exhibit characteristics that may persist even in the single-particle picture, particularly under the extreme condition of U=0, accompanied by a substantial Zeeman splitting. It would be valuable for the authors to provide a more detailed explanation and elaboration on this aspect.
-
Revisiting the Local conductance presented in Fig. 3 using standard transport methods, such as Non-Equilibrium Green's Function (NEGF), holds the potential to offer valuable insights, particularly in light of the absence of interactions in the outer dots, as mentioned in the previous comment regarding U=0.
-
The formulation related to Local conductance in Fig. 3 needs detailed explanation, which was unfortunately omitted in the current version.
-
The introduction of a coupling strength $\Gamma_{LD}=0.01$ (Fig. 3) requires clarification in comparison with the tunneling Hamiltonian defined in Eq. 4.
-
Discrepancies between the description of Majorna wave functions in Fig. 3-d and the text. It appears that the wavefunctions for cases with $\mu_M=0.2$ and $\mu_M=0.5$ are differently interpreted in the figure and the accompanying explanation, necessitating clarification.
Taking into account the aforementioned comments, I suggest that the manuscript could be suitable for publication in SciPost Phys. Core rather than SciPost Phys, provided that the authors address the outlined concerns and make necessary revisions to enhance clarity and coherence.

---

## Round 1 · Referee Report · Anonymous (Referee 2) · 2024-3-26

Report
In this work the authors present an interesting idea of using a Josephson junctions to tune the coupling between two quantum dots to create a Kitaev chain. The advantage of the setup proposed is that the coupling between the quantum dots can be controlled by tuning the flux threading the Josephson junction allowing to tune via the flux the topological character of the chain.
The work is well written and is innovative and I find that it could be published in SciPost Physics.
There are few items that I think the authors should address in a revised version of the manuscript.
-
It would be helpful if in Fig. 1 the authors would also include a schematic of the real system giving a sense of its geometry.
-
A more substantial question relates to the type of Josephson junction needed to realize the setup. In the text the authors mention that they "use a planar Josephson junction". Is this necessary? Must the Josephson junction be very wide and therefore planar? Or the requirement is simply that the Junction must be large enough to be able to tune the flux without having to use very large magnetic fields? What about the length of the junction? Does it have to be short?
-
On page 4 Delta_CAR and Delta_ECT are expressed in terms of the coherence factors u and v, but these are not defined in the text.
-
Units should be added for all the axes of all the panels of Fig.3.
-
Given the order of the discussion in the main text the columns of Fig. 3 should be switched.
-
A couple of typos (for instance has->is 3rd line on page 7).

---

## Round 1 · Referee Report · Anonymous (Referee 3) · 2024-4-9

Strengths
- In light of recent experimental developments with quantum-dot-based short Kitaev chains, the setup described in this work seems experimentally feasible and interesting for the Majorana community.
- The manuscript is well written.
Weaknesses
-
I do not see significant advancements in this work in terms of originality and methodology.
-
The authors do not articulate which of the 4 "expectations", required for SciPost Physics publications, are met by their work.
See further comments in "Report".
Report
The authors study a model of a two-site Kitaev chain, where the two sites are single-orbital quantum dots with large Zeeman splitting and large on-site Coulomb repulsion, whereas tunnel coupling is mediated by a third dot embedded in a Josephson junction. A key finding, illustrated in Fig. 2a,c, is that by tuning the flux bias of the Josephson junction, the “topological dimerized limit” (aka “sweet spot”) of the two-site Kitaev chain can be reached. The authors also discuss “optimal sweet spots” in the parameter space, where the even-odd degeneracy of the sweet spot is robust against parameter fluctuations.
In light of recent experimental developments with quantum-dot-based short Kitaev chains, the setup described in this work seems experimentally feasible and interesting for the Majorana community. Also, the manuscript is well written. I have to add that I do not see significant advancements in this work in terms of originality and methodology.
My further comments are as follows. In my view, the authors reaction to the first 3 comments will be important to judge if this work should be published in SciPost Physics (cf. https://scipost.org/SciPostPhys/about#criteria).
1) The authors do not articulate which of the 4 "expectations", required for SciPost Physics publications, are met by their work. As a working hypothesis, I assume that they think that their work opens “a new pathway in an existing or a new research direction, with clear potential for multipronged follow-up work;”
2) I do understand the conceptual framework of the Kitaev chain, and how a realisation of it could lead to braiding-based robust control of Majorana qubits, and their protection against certain noise types. However, my understanding is that these features arise only in the non-interacting case. The authors study quantum dots with a large Hubbard-U interaction, i.e., away from the non-interacting case. Therefore I think the authors should discuss if their setup is still relevant for building protected Majorana qubits, and if it is, then how? If it turns out that they cannot justify the relevance of this setup for building protected Majorana qubits, then I doubt that the expectation I mentioned in 1) is fulfilled.
3) To my understanding, the robustness of a Majorana qubit against decoherence requires long chains. The authors present a two-site chain. Is their idea of using a flux-biased Josephson junction scalable to longer chains? If it is not, then this should be clearly discussed. If it is, then this should also be detailed, preferably with an image showing the geometry of a scaled-up device. Again, the response here affects how well the expectation I mentioned in 1) is fulfilled.
Further comments for the authors' consideration:
4) The term “sweet spot” is used in the text in different roles. (i) It refers to the “t = Delta” limit (fully dimerized limit) of the Kitaev chain, (ii) In Fig. 3 caption (and also in the text), it is also used as a parameter point (black cross) where the flux value needed to maintain the even-odd degeneracy is first-order insensitive to the common on-site energy of the outer dots, (iii) at the bottom of page 6, “sweet spot” is used to describe parameter points where the Majorana energy is insensitive to flux fluctuations. I find this confusing, and hence recommend to clarify the terminology, e.g., introduce different names for the different sweet spots (or at least use different qualifiers before “sweet spot”) .
5) In the abstract, the authors mention that they consider a “narrow” Josephson junction, but in the text they do not explain what that means and why narrowness is important. I recommend to either omit “narrow” or explain it.
6) In the introduction, the authors write that “The flux control method eliminates challenges associated with charge noise…”. Why is that true? I think charge noise is present in a gate-control quantum dot array, no matter if flux control or voltage control is used.
7) I recommend to the authors that they make an attempt to state the range of validity of the Hamiltonian defined in Eqs. (1)-(4), with the goal to allow experimentalists to judge whether their experimental setup is well described by this minimal Hamiltonian or not. E.g., I presume that Kondo-type correlations are not captured by this Hamiltonian, etc. Also, I recommend to comment on the feature that the middle dot has no Coulomb repulsion whereas the outer dots have strong Coulomb repulsion: what are the physical conditions in a device to achieve this setting, at least approximately?
8) The text says “As depicted in Fig. 3(d), the wavefunctions of the two Majoranas are completely localized on the left and right dots, respectively”, but in Fig. 3(d) the majority of the wavefunction is on the middle dot. I recommend to resolve this contradiction.

---

## Round 2 · Referee Report · Anonymous (Referee 4) · 2024-7-31

Report

I would like to thank the authors for their efforts in responding to the comments and significantly improving the text. The manuscript is now much clearer and better organised.

In the theory developed by the authors, it is still not clear what role "U" plays. The authors simply set U=2 and present their results without giving details of its effects. As I mentioned in my previous comment, if "U" does not play a significant role in this setup, why do the authors insist on considering a many-body problem instead of working within single-particle physics?

Moreover, the benchmark of ED and the effective Hamiltonian added in the revised version shows that for parameters $ t \approx 0.1 \ll E_Z=4 $ this effective theory cannot accurately capture the physics close to phase $ \phi=\pi $ (see Fig. 7 in the Appendix). In addition, the effect of changing "t" by an order of magnitude from 0.01 to 0.1 was not discussed in the main text.

In their response, the authors suggested that their work could potentially be seen as "opening a new avenue in an existing or emerging research direction, with clear potential for multi-pronged follow-up work". However, as I stated in my previous review, I think this paper is better suited to the SciPost Physics Core critique as stated on their website.

Recommendation

Accept in alternative Journal (see Report)

---

## Round 2 · Referee Report · Anonymous (Referee 5) · 2024-8-11

Report

In their reply and revised version the authors addressed all the concerns I had reaised in my first report. There are only a couple of things that the authors could add to make the work more self-contained:
1. Explain the concept of "poor man Majorana"
2. In Majorana nanowires zero energy (in the ideal setup and no disorder) is equivalent to large spatial separation of the Majorana modes and topological protection, but for the proposed setup this doesn't seem to be the case. It would be helpful if the authors could explain the relation between zero for the energy eigenvalue, topological protection, and overlap of Majorana wave functions to more clearly address question 3 of Referee 3.

I find this paper to be a nice addition to the series of works proposing to use chains of quantum dots to create Majoranas. The authors propose a neat way to tune the chain in the topological regime. The work focuses on a specific aspect of the proposal and so it might fit better in SciPostPhysCore than SciPostPhys.

Recommendation

Publish (easily meets expectations and criteria for this Journal; among top 50%)

---

## Round 2 · Referee Report · Anonymous (Referee 6) · 2024-8-19

Report

I thank the authors for all their responses and revisions.

My key remarks were 2 -- the role of the Hubbard U, and 3 -- scaling up for longer chains.

As for 3), I find the newly added Fig. 5 insightful. Naturally, this figure also brings up many new questions, e.g., how to size the flux loops; how to fit all gates, flux loops and their flux control lines on a chip, etc. But on a conceptual level, it nicely illustrates the opportunity of scaling.

The authors also address my comment 2). I do find the response oversimplified. I do understand that, as the authors write, "in the large Zeeman limit opposite spin orbital and doubly occupied states can be neglected", although it would be more accurate to write "in the infinite Zeeman limit". However, the authors go further, and write "braiding quality does not depend on interactions since the many-body model with large $U$ and $E_Z$ is equivalent to a single particle model with large $E_Z$ and $U=0$". I find this unjustified. In fact, my impression is that it is a *critical open question* of the field how braiding quality is affected by a large but non-infinite Zeeman splitting and a large but non-infinite Coulomb repulsion. It is well conceivable that the quantised pi/2 phase gate obtained in the noninteracting (or U->infinity) case is modified as U is decreased to a finite value, and that could imply a strong obstruction to the quantum-dot based path toward Majorana qubits, pursued in this work.

I do acknowledge the merits of this work as I described in my previous report. However, based on the above considerations, I am reluctant to see that this work opens "a new pathway in an existing or a new research direction, with clear potential for multipronged follow-up work", and hence I recommend acceptance in SciPost Physics Core.

Recommendation

Accept in alternative Journal (see Report)

---

## Round 2 · Author Response

We thank the referees for their comments on the manuscript. Here we provide a detailed response to the comments of each referee.

Referee 1

  1. Inconsistency between Fig. 1-b and the Hamiltonian defined in Eq. 3, Eq. 5, or Eq. 8. The schematic in Fig. 1-b, particularly the orange representation, appears to deviate from the described Hamiltonian formulations.

We thank the referee for pointing out this inconsistency. We would like to clarify that the orange dots in Fig. 1-b represent the Andreev quasiparticles in the excitation basis. In order to clarify this point we have expanded the text about the excitation basis, and written down the explicit states in the figure.

  1. Ambiguities in Fig. 2. It would enhance clarity if the authors compared the energy spectrum and electron-hole components (u, v) of Eq. 5 with the effective Hamiltonian (Eq. 8) and included these comparisons in panels a and b, respectively. Currently, panels a and b seem to derive from Eq. 5, while panels c and d derive from the effective Hamiltonian, which could be confusing for readers.

We thank the referee for pointing out this possible source of confusion. Indeed, the results shown in the upper row of Fig. 2 were obtained using the model with ABS whereas the results shown in the lower row panel were obtained using the effective model. We have clarified this difference in the caption of Fig. 2.

  1. The effective Hamiltonian (Eq. 8) should be elaborated with details upon to ensure self-consistency, especially considering its role in benchmarking with the numerical Exact Diagonalization (ED) results.

We have added an appendix (Appendix B) where we benchmark the CAR and ECT terms in the effective Hamiltonian with ED results. We also discuss the regime of parameters where they are valid as well as their deviations.

  1. The theory developed by the authors regarding the limit of large U in the outer dots appears to exhibit characteristics that may persist even in the single-particle picture, particularly under the extreme condition of U=0, accompanied by a substantial Zeeman splitting. It would be valuable for the authors to provide a more detailed explanation and elaboration on this aspect.

The referee is right by pointing out that the limit of large $E_Z$ and $U=0$ is equivalent to our case and described by single-particle physics. We include finite $U$ in the quantum dots because that is closer to what is observed in experiments. We have added a footnote, footnote 1, about this point in the appendix: "The behaviour of the system in the large charging energy limit would be similar to the case with vanishing charging energy given that the Zeeman splitting in the quantum dots is sufficiently large."

  1. Revisiting the Local conductance presented in Fig. 3 using standard transport methods, such as Non-Equilibrium Green's Function (NEGF), holds the potential to offer valuable insights, particularly in light of the absence of interactions in the outer dots, as mentioned in the previous comment regarding U=0.

We thank the referee for their comment. Since our proposal features quantum dots which feature large charging energy $U$, the case $U=0$ is not relevant. Secondly, even though large charging energy doesn't affect the ground state properties in the spinless regime, it may have an affect on the excited states, which we show in our Figure 3(e-f).

  1. The formulation related to Local conductance in Fig. 3 needs detailed explanation, which was unfortunately omitted in the current version.

We thank the referee for pointing this out. We indeed omitted the explanation of the method used to calculate the local conductance. We use the rate equation which was described in detail in Phys. Rev. B 106, L201404 (2022) (our Ref 45 in the current version). We now mention the method and describe it in the text of the revised version.

  1. The introduction of a coupling strength $\Gamma_{LD} = 0.1$ (Fig. 3) requires clarification in comparison with the tunneling Hamiltonian defined in Eq. 4.

We point out to the referee that this is a parameter that describes the coupling of the outer dots to an external lead, and it is required for the rate equation calculation. To avoid confusion, we have added the following sentence to our manuscript: "In particular, we consider leads at a finite temeperature $T$ coupled to the outer dots with a coupling strength $\Gamma_{LD}$."

  1. Discrepancies between the description of Majorna wave functions in Fig. 3-d and the text. It appears that the wavefunctions for cases with $\mu_M=0.2$ and $\mu_M=0.5$ are differently interpreted in the figure and the accompanying explanation, necessitating clarification.

We thank the referee for pointing out a possible source of confusion. We are indeed refering to Fig. 3d and to clarify it, we have replaced the sentence: "As depicted in Fig. 3(d), the wavefunctions of the two Majoranas are completely localized on the left and right dots, respectively." For the sentence: "As depicted in Fig. 3(d), the wavefunctions of the two Majoranas are decoupled. Namely, the first Majorana wavefunction is localized in the left and the second one in the right."

Referee 2

  1. It would be helpful if in Fig. 1 the authors would also include a schematic of the real system giving a sense of its geometry.

We thank the referee for their comment. We have included a figure, Figure 5 in the current version, of the proposed geometry for a longer Kitaev chain with flux tunability.

  1. A more substantial question relates to the type of Josephson junction needed to realize the setup. In the text the authors mention that they "use a planar Josephson junction". Is this necessary? Must the Josephson junction be very wide and therefore planar? Or the requirement is simply that the Junction must be large enough to be able to tune the flux without having to use very large magnetic fields? What about the length of the junction? Does it have to be short?

We thank the referee for their question. The only requirement for a Josephson junction is to be a short junction with a single ABS. To clarify this point, we adopted the term "short junction" in the revised manuscript.

  1. On page 4 Delta_CAR and Delta_ECT are expressed in terms of the coherence factors u and v, but these are not defined in the text.

We thank the referee for pointing this out. We have defined the coherence factors in the text in Eq. 7 of the revised manuscript.

  1. Units should be added for all the axes of all the panels of Fig.3.

We thank the referee for their comment. In the manuscript below Eq. 6, we set the overall energy scale to be $\Gamma_+$ and we set it to be unity, i.e. $\Gamma_+=1$. Based on this choice, all other parameters are in units of $\Gamma_+$.

  1. Given the order of the discussion in the main text the columns of Fig. 3 should be switched.

We thank the referee for pointing this out. Following their suggestion, we switched the columns of Fig. 3.

  1. A couple of typos (for instance has->is 3rd line on page 7).

We thank the referee for pointing out the typos. We have fixed them in our revised version of the manuscript.

Referee 3

1) The authors do not articulate which of the 4 "expectations", required for SciPost Physics publications, are met by their work. As a working hypothesis, I assume that they think that their work opens “a new pathway in an existing or a new research direction, with clear potential for multipronged follow-up work;”

We agree with the referee that our work opens a new pathway in an existing or a new research direction because: * it uses a fundamental property of Andreev bound states which hasn't been explored before in this context, * it is experimentally realisable with state-of-the-art Josephson junctions, * it is straightforward to scale up for larger chains, * it provides advantages in the experimental control of PMM properties.

2) I do understand the conceptual framework of the Kitaev chain, and how a realisation of it could lead to braiding-based robust control of Majorana qubits, and their protection against certain noise types. However, my understanding is that these features arise only in the non-interacting case. The authors study quantum dots with a large Hubbard-U interaction, i.e., away from the non-interacting case. Therefore I think the authors should discuss if their setup is still relevant for building protected Majorana qubits, and if it is, then how? If it turns out that they cannot justify the relevance of this setup for building protected Majorana qubits, then I doubt that the expectation I mentioned in 1) is fulfilled.

We thank the referee for their comment. We point out to the referee that in the large Zeeman limit opposite spin orbital and doubly occupied states can be neglected. Therefore, the low-energy physics of this model can be described by a non-interacting model with Majorana bound states. In recent papers regarding Majorana qubits in quantum dot chains with high charging energy, such as PRX Quantum 5, 010323 (2024) and Phys. Rev. B 108, 085437 (2023), braiding quality is characterised by the properties of the low-energy degrees of freedom. Furthermore, as pointed out by referee 1, braiding quality does not depend on interactions since the many-body model with large $U$ and $E_Z$ is equivalent to a single particle model with large $E_Z$ and $U=0$.

3) To my understanding, the robustness of a Majorana qubit against decoherence requires long chains. The authors present a two-site chain. Is their idea of using a flux-biased Josephson junction scalable to longer chains? If it is not, then this should be clearly discussed. If it is, then this should also be detailed, preferably with an image showing the geometry of a scaled-up device. Again, the response here affects how well the expectation I mentioned in 1) is fulfilled.

We thank the referee for their comment. Indeed, the topological protection holds only for long chains and our design allows for scaling up the chain. Upon referee's suggestion, we have now included a device design, Figure 5, for a longer flux-tunable Kitaev chain.

4) The term “sweet spot” is used in the text in different roles. (i) It refers to the “t = Delta” limit (fully dimerized limit) of the Kitaev chain, (ii) In Fig. 3 caption (and also in the text), it is also used as a parameter point (black cross) where the flux value needed to maintain the even-odd degeneracy is first-order insensitive to the common on-site energy of the outer dots, (iii) at the bottom of page 6, “sweet spot” is used to describe parameter points where the Majorana energy is insensitive to flux fluctuations. I find this confusing, and hence recommend to clarify the terminology, e.g., introduce different names for the different sweet spots (or at least use different qualifiers before “sweet spot”).

We thank the referee for pointing out this source of confusion about the terminology of sweet spot. First of all, points (i) and (ii) raised by the referee are equivalent because the bottom of the degeneracy line corresponds to the point where CAR is equal to ECT. In the perturbative regime, this point exactly corresponds to "t=Delta". About point (iii) raised by the referee, we have extended the discussion in the main text referring to Figure 4 as follows: "In Fig.~\ref{fig:phase_dependence} every point corresponds to the sweet spot identified at the bottom of the degeneracy line in the phase diagram as in Fig.\ref{fig:phase_diagram}(a, d). It is not always possible to tune into a sweet spot because when the superconducting phase is around $\pi$ for a nearly symmetric Josephson junction, the ABS becomes gapless, making the physical mechanism of using ABS as a coupler break down. Nevertheless, outside those regions in parameter space, we are able to characterize the quality of the Majorana excitations by computing the variation of the ground state energy splitting $E_M \equiv E_{\text{odd,gs}} - E_{\text{even,gs}} $ with respect to phase $\phi$ and the excitation gap $E_\text{gap}$."

5) In the abstract, the authors mention that they consider a “narrow” Josephson junction, but in the text they do not explain what that means and why narrowness is important. I recommend to either omit “narrow” or explain it.

We thank the referee for their comment. The only requirement for a Josephson junction is to be a short junction with a single ABS. To clarify this point, we adopted the term "short junction" in the revised manuscript.

6) In the introduction, the authors write that “The flux control method eliminates challenges associated with charge noise…”. Why is that true? I think charge noise is present in a gate-control quantum dot array, no matter if flux control or voltage control is used.

We thank the referee for their question. Here, we mean that we can tune the system to a sweet spot using flux knob, without changing the electrostatic environment. Changing the electrostatic environment using electrostatic gates may induce charge noise and cross-talk, that is undesirable. To clarify this point further, we have modified that sentence to "The flux control method eliminates challenges associated with cross-talk often encountered in electrostatic gate voltage control."

7) I recommend to the authors that they make an attempt to state the range of validity of the Hamiltonian defined in Eqs. (1)-(4), with the goal to allow experimentalists to judge whether their experimental setup is well described by this minimal Hamiltonian or not. E.g., I presume that Kondo-type correlations are not captured by this Hamiltonian, etc. Also, I recommend to comment on the feature that the middle dot has no Coulomb repulsion whereas the outer dots have strong Coulomb repulsion: what are the physical conditions in a device to achieve this setting, at least approximately?

The model Hamiltonians in Eq.1-4 have been extensively studied theoretically for Kitaev chain physics in quantum-dot-superconductor arrays [Phys. Rev. B 106, L201404 (2022), arXiv:2310.09106 and Phys. Rev. Research 5, 043182 (2023)]. The corresponding model simulations show excellent agreement with experimental measurements in tunnel spectroscopy [Nature 630, 329–334 (2024), arXiv:2311.03193]. Thus we believe that it is well justified that the minimal Hamiltonian provides a very good description for the experimental devices under study. On the other hand, hybrid regions (hosting Andreev bound states) usually have a negligible charging energy because of strong coupling to a grounded metallic supercondutor, while normal dots tend to have a large one since they are only tunnel coupled to superconductor or normal leads.

8) The text says “As depicted in Fig. 3(d), the wavefunctions of the two Majoranas are completely localized on the left and right dots, respectively”, but in Fig. 3(d) the majority of the wavefunction is on the middle dot. I recommend to resolve this contradiction.

We thank the referee for pointing out a possible source of confusion. Here, we mean that the Majoranas remain completely decoupled on the left and right dots respectively, even though the weight of their wavefunctions increases on the middle dot. To clarify this point, we have added the following sentences to our manuscript: "As depicted in Fig. 3(b), the wavefunctions of the two Majoranas are decoupled. Namely, the first Majorana wavefunction is localized in the left and the second one in the right."

---

## Round 2 · List of Changes

We provide a version that highlights the changes in the manuscript we have made publicly available in the following link: https://surfdrive.surf.nl/files/index.php/s/drXeNxbO11YAmbk

---

## Editorial Decision

published